# Paramyxoviruses in Old World fruit bats (Pteropodidae): An open database and synthesis of sampling effort, viral positivity, and coevolution

Maya M. Juman[1]*, Olivier Restif[1], Daniel J. Becker[2]

**1** Department of Veterinary Medicine, University of Cambridge, Cambridge, United Kingdom, **2** School of Biological Sciences, University of Oklahoma, Norman, Oklahoma, United States of America

* mmj38@cam.ac.uk

## Abstract

Over the past 30 years, research interest in the links between Old World fruit bats (family Pteropodidae) and paramyxoviruses has driven a rapid proliferation of surveillance studies in this system. We performed a systematic review and data synthesis of all published paramyxovirus studies of wild pteropodids. Here, we present an open, static, PRISMA-compliant database called pteroparamyxo, which includes 1,476 records of prevalence or seroprevalence from 118 studies published between 1971 and 2023. Using this dataset, we examined biases in sampling effort, predictors of viral positivity, tissue tropism, and pteropodid–paramyxovirus coevolution. We found significant spatial and taxonomic bias in sampling effort, largely driven by overrepresentation of *Pteropus*–henipavirus systems; in particular, African bats were undersampled for paramyxoviruses relative to Asian and Oceanian taxa. On the viral side, henipaviruses (specifically Hendra, Nipah, and Cedar viruses) were overrepresented relative to pararubulaviruses. We also identified substantial non-random variability in paramyxovirus prevalence and seroprevalence. Spleens and pooled fecal samples from pteropodids yielded the highest PCR positivity, while samples from Oceania tended to have slightly higher seroprevalence than those from Asia and Africa. Collection year was not a significant predictor of positivity, suggesting limited overall change in paramyxovirus PCR prevalence or seroprevalence over the last several decades. Finally, we found weak evidence of pteropodid–paramyxovirus coevolution, supporting relatively frequent host-switching within this system. Our study highlights critical sampling gaps to address in future surveillance studies and provides preliminary evidence for sample and tissue types to prioritize in field- and museum-based sampling.

**Data availability statement:** All relevant data are within the manuscript and its Supporting Information files and https://github.com/mayajuman/pteroparamyxo.

**Funding:** MMJ was supported by a Gates Cambridge Scholarship enabled by grant OPP1144 from the Bill & Melinda Gates Foundation (https://www.gatescambridge.org/) as well as the Fellow-in-Residence award from the Viral Emergence Research Initiative (Verena) funded by NSF DBI 2515340 (https://www.viralemergence.org/). DJB was supported by NSF DBI 2515340 and the Edward Mallinckrodt, Jr. Foundation (https://emallinckrodtfoundation.org/home-page). No funders played a role in the study design, data collection and analysis, decision to publish, or preparation of the manuscript.

**Competing interests:** The authors have declared that no competing interests exist.

## Author summary

Paramyxoviruses are a family of viruses that include the human measles and mumps viruses as well as emerging zoonoses like Hendra and Nipah henipaviruses. These henipaviruses spill over from wild fruit bats to domestic animals and humans in Australia and Asia, causing high fatality rates with no known human treatments or vaccines. The occurrence and prevalence of paramyxoviruses across the entire fruit bat family (Pteropodidae) has never been formally investigated, as published detection attempts have not yet been consolidated into a database. We conducted a systematic review of the existing literature on pteropodid–paramyxovirus sampling efforts and compiled an open database for researchers to reference and use. We then leveraged this novel database for a suite of analyses and demonstrated that Australian and Asian fruit bats are disproportionately sampled for paramyxoviruses while African species are understudied, despite the fact that African samples have slightly higher rates of paramyxovirus detection. We identified sample types that may be particularly likely to yield viral detections, a finding which can help guide future survey efforts using field studies as well as museum collections. We also demonstrate that paramyxoviruses appear liable to switch host species within Pteropodidae, challenging earlier hypotheses that these viruses are host specific.

## Introduction

Anthropogenic change is driving increased viral transmission across species boundaries, including spillover of zoonotic viruses from wildlife to humans [1]. One particular viral family of interest is the *Paramyxoviridae*, negative-strand RNA viruses identified across vertebrate groups. Paramyxoviruses cause particularly high disease burdens in mammals and birds, with varying degrees of host specificity [2]. For example, the subfamily *Orthoparamyxovirinae* contains the genera *Morbillivirus* (e.g., host-specific measles virus and broadly infecting canine distemper virus) and *Henipavirus* (e.g., zoonotic bat-borne Hendra and Nipah viruses, which cause fatal encephalitis in horses and humans) [3]. The subfamily *Rubulavirinae* contains the genera *Orthorubulavirus* (e.g., mumps and human parainfluenza viruses) and *Pararubulavirus* (e.g., Tioman virus, Menangle virus, and other recently isolated bat-borne viruses) [3]. To understand the host range, geographic range, and potential spillover risks of paramyxoviruses, a more comprehensive picture of host–paramyxovirus relationships is critical.

Bats are known to host a variety of paramyxoviruses and have been suggested as the ancestral hosts of *Paramyxoviridae* [4]. In particular, Old World fruit bats (family Pteropodidae) have been implicated as reservoir hosts of two highly pathogenic henipaviruses and other potentially zoonotic paramyxoviruses. This ecologically and morphologically diverse bat family of roughly 200 species is widespread across Asia, Oceania, and Africa [5]. Immunity-related genes in pteropodids have evolved at

higher rates relative to other chiropteran families, and some pteropodid-specific mutations are associated with decreased inflammatory responses to pathogens [6]. A balance of viral tolerance and immune defense strategies may explain these fruit bats' apparent ability to host viruses that are highly pathogenic in other mammals [7]. Henipaviruses, including Hendra virus [8], Nipah virus [9], Ghanaian bat henipavirus [10], and Cedar virus [11], have been detected in pteropodids. Novel pararubulaviruses have also been isolated from pteropodids [12–15]. Some studies have noted apparent clustering of related paramyxoviruses in certain pteropodid species, raising the possibility of host specificity and host-virus coevolution [16], though this has not been quantitatively tested. This growing body of evidence linking pteropodids to paramyxoviruses has also prompted further research on the ecological links between these bats and their viruses [17–20], which have implications for downstream spillover risk [21,22].

Pteropodids are largely frugivorous and nectarivorous, and loss of native habitat has driven many of these species to seek food sources in urban areas (e.g., public and private gardens), increasing the likelihood of human–bat interactions [5]. Climate and land-use change have been linked to the spillover of henipaviruses from pteropodids [22,23], and the geographic (and possibly host) ranges of paramyxoviruses appear to be expanding in certain cases—for example, the recent Nipah virus spillovers and associated fatalities in South India [24,25]. The resulting interest in henipaviruses may have created a bias toward studying *Pteropus* bats in Australia and South and Southeast Asia [23]. Sampling disparities such as these, coupled with a lack of consolidated data, prevent a deeper understanding of global changes in viral occurrence, distributions, and spillover rates over time [26]. Large, taxonomically broad host–virus association databases partially address this issue, but these presence-only databases lack negative results by nature and also often lack fine-grained metadata. While taxonomically specific databases exist for other viral families like coronaviruses [27], interactions between pteropodids and the broader *Paramyxoviridae* have not been comprehensively compiled, despite the fact that these bats have been heavily sampled for henipaviruses and other paramyxoviruses [28–30]. The absence of such a synthesis limits the identification of patterns in viral prevalence, which is critical for spillover prediction and mitigation [31].

To address these gaps and enable formal synthesis of pteropodid–paramyxovirus associations, we built a standardized, PRISMA (Preferred Reporting Items for Systematic Reviews and Meta-Analyses)-compliant database of all published detection attempts of paramyxoviruses in wild pteropodid bats, including metadata on viral taxonomy, host taxonomy, sampling and detection methodologies, seasonality, location, and date, as well as including data on negative results. We then used the resulting dataset to perform a series of phylogenetic comparative analyses and formal meta-analyses to better understand pteropodid–paramyxovirus relationships and the current state of the associated literature. To this end, we investigated: (1) geographic and taxonomic (both host and viral) sampling gaps in this system; (2) key predictors of paramyxovirus prevalence and seroprevalence in pteropodids, including potential tissue tropism; and (3) the degree of pteropodid–paramyxovirus coevolution or host-switching.

## Methods

### Systematic review

To generate our database of paramyxovirus detection attempts in wild pteropodids, we followed the PRISMA protocol (S1 Fig) [32]. On August 3rd, 2023, we systematically searched PubMed and Web of Science with the following string: (pteropodid* OR pteropid* OR "fruit bat" OR "fruit bats" OR "flying fox" OR "flying foxes" OR *Acerodon* OR *Aethalops* OR *Alionycteris* OR *Aproteles* OR *Balionycteris* OR *Boneia* OR *Casinycteris* OR *Chironax* OR *Cynopterus* OR *Desmalopex* OR *Dobsonia* OR *Dyacopterus* OR *Eidolon* OR *Eonycteris* OR *Epomophorus* OR *Epomops* OR *Haplonycteris* OR *Harpyionycteris* OR *Hypsignathus* OR *Latidens* OR *Macroglossus* OR *Megaerops* OR *Megaloglossus* OR *Melonycteris* OR *Mirimiri* OR *Myonycteris* OR *Nanonycteris* OR *Neopteryx* OR *Nesonycteris* OR *Notopteris* OR *Nyctimene* OR *Otopteropus* OR *Paranyctimene* OR *Penthetor* OR *Pilonycteris* OR *Plerotes* OR *Ptenochirus* OR *Pteralopex* OR *Pteropus* OR *Rousettus* OR *Scotonycteris* OR *Sphaerias* OR *Stenonycteris* OR *Styloctenium* OR *Syconycteris* OR *Thoopterus*) AND

(paramyxovirus* OR *Paramyxoviridae* OR henipavirus* OR Hendra OR Nipah OR rubulavirus* OR morbillivirus*). We identified 1,260 records in total and proceeded to screen 862 titles and abstracts from non-duplicated records. Studies were excluded if they did not include new (i.e., not republished) data on paramyxovirus detection attempts in wild bats—therefore, any studies of captive bats or experimental studies of bat-derived cell lines were excluded. We identified 136 articles for further full-text screening, of which 118 met the inclusion criteria for our study.

## Data collection

Our final database included 118 studies and 1,476 records. Where prevalence was not reported in the original studies, it was calculated from the number of positive and tested samples before meta-analysis. For viral isolation attempts where positivity rate or number of positive samples was not reported, we included a binary variable (0: unsuccessful detection; 1: successful detection). Therefore, the same bat may be included in multiple records, if the study tested multiple samples (e.g., different tissue types or detection methods) per individual bat. For papers where over 100 sampling events are detailed, sampling events were summarized by location, species, and tissue type. When original studies did not separate samples by species, locations, tissue types, or years, all information was included in one record (separated by commas in the relevant cells). Before downstream analyses, these pooled records were filtered out when relevant, but they remain in the database for researchers who may be interested in specific species, tissue types, or sites. When original studies did not report all the metadata for a study (e.g., sampling season), these were not imputed.

## Spatial and taxonomic sampling biases

Our dataset included bat samples from 33 countries in three geographical regions (Africa, Asia, and Oceania; countries classified according to the United Nations Geoscheme). All analyses were conducted in R (version 4.2.1) [33]. We first visualized the number of studies and (log-scale) samples per country by building maps with the aggregated Pteropodidae IUCN range map overlaid for context. We then fitted three GLMs to these country-level variables with region as the categorical predictor and one of three respective response variables: binary sampling status (whether or not a country had been sampled for paramyxoviruses in pteropodids) as a binomial response, with all countries in these three regions included; the number of unique studies per country as a Poisson-distributed response, with only sampled countries included; and the number of samples per country as another Poisson-distributed response, with only sampled countries included. We assessed the fit of GLMs using McFadden's $R^2$ ('performance' package) and made individual region-to-region pairwise comparisons using the 'emmeans' package [34,35].

To assess taxonomic sampling biases, we first aligned all sampled bats ($n = 58$) and viruses ($n = 16$) identified at the species level to host [36] and virus [3] phylogenies. To do so, we relabeled species names to fit the published taxonomy where necessary (e.g., *Pteropus medius* was relabeled as *P. giganteus* for the purpose of taxonomic backbone alignment, though we retain the currently recognized names in our published database). For all bat and virus species in the phylogenies, we created a binary response variable representing whether or not each species had been sampled or sampled for, respectively. For those species present in our dataset, we calculated the number of unique studies and total samples associated with each bat and virus species. To assess the degree of phylogenetic signal in sampling effort across bats and viruses, we used the 'caper' package [37] to calculate $D$ (<0: highly conserved; 0: phylogenetic clustering under Brownian motion; 0–1: intermediate phylogenetic signal; 1: phylogenetically random sampling distribution; >1: overdispersed) for binary sampling effort and Pagel's lambda ($\lambda$) (0: phylogenetically random sampling distribution; 0–1: intermediate phylogenetic signal; 1: phylogenetic clustering under Brownian motion) for the $\log_{10}$-transformed number of studies and samples. Finally, we used the package 'phylofactor' [38] to apply phylogenetic factorization, a graph-partitioning algorithm, to both the bat and virus phylogenies. This approach partitioned binary sampling effort (binomial response), number of studies (Poisson distribution), and number of samples (Gaussian distribution due to an extreme skew in number of

samples and consequent transformation via natural log) across the trees using iterative GLMs, producing a list of bat or virus clades (containing at least three species) that differ significantly from the rest of the tree in sampling effort. We used Holm's sequentially rejective test with a family-wise error rate of 5% to account for multiple comparisons and determine the number of statistically significant clades [39].

## Meta-analysis of viral and antibody positivity

We used the 'metafor' package [40] to run two hierarchical meta-analysis models for two separate viral positivity datasets: PCR detection ($n = 782$ records) and serological detection ($n = 311$ records). The response variables in our models were the Freeman-Tukey double arcsine-transformed proportions of paramyxovirus PCR-positive or seropositive samples and their sampling variances. Models were fitted using restricted maximum likelihood (REML) and included bat genus, genus-level phylogeny [36], and country as random effects along with an observation-level random effect nested inside a study-level random effect; all models were also weighted by inverse sampling variance. We used genus instead of species as a random effect in these models to retain records from bats not identified to the species level (e.g., unspecified Australian *Pteropus* spp.). We calculated the contribution of heterogeneity (true variability rather than noise) to variance in viral positivity ($I^2$) for the entire dataset, as well as the proportional contribution for each random effect. Our models included all available non-collinear predictors. For the PCR model, we included fixed effects of region (Oceania, Asia, Africa), study type (cross sectional, longitudinal, or both), sample type (feces, serum, saliva, swab, tissue, various), sampling year, and host subfamily (Cynopterinae, Eidolinae, Epomophorinae, Macroglossusinae, Pteropodinae, Rousettinae); for the serology model, we included fixed effects of region (Oceania, Asia, Africa), sampling year, and host subfamily (Cynopterinae, Eidolinae, Epomophorinae, Macroglossusinae, Pteropodinae, Rousettinae). We note that seroprevalence is determined with different methods and cutoff values across the studies and that serological assays also capture cross-reactivity with different antigens; here, we seek to understand broad patterns in general paramyxovirus seroprevalence. For records reflecting detection attempts across a range of sampling years, the midpoint of the start and end years was used to calculate the average "sampling year". Methodological variation within datasets could not be included in these models due to collinearity with other predictors; however, changes in diagnostic methods track with time (e.g., a shift from RT-PCR to metagenomic methods for PCR detection and a shift from neutralization assays to ELISA and multiplexed assays for serological detection), such that including sampling year as a predictor accounts for this variation (S2 Fig). Limited temporal autocorrelation in both PCR prevalence and seroprevalence also suggests that sampling year effectively controls for any temporal clustering of studies (S3 Fig).

To assess paramyxovirus prevalence in different tissue types (as tissue type was a collinear predictor in all possible models and therefore could not be included), we also calculated overall viral prevalence per tissue type for records using PCR.

## Cophylogenetic analysis

We lastly used the Procrustean Approach to Cophylogeny (PACo), with the 'paco' package, to assess coevolution between paramyxoviruses and pteropodids and thus the dependence of the viral phylogeny on the host phylogeny [41]. For this analysis, we defined a known association as either a PCR detection or viral isolation of an ICTV-recognized paramyxovirus species in a described species of pteropodid, thereby excluding serological detections (due to lower viral specificity) and any PCR or viral culture detection in bats recognized only to the genus level (e.g., unspecified Australian *Pteropus* spp.). We conducted a goodness-of-fit test to evaluate how well a codivergence model fits the matrix of known host–virus links, and then established its statistical significance by fitting the model to 5000 randomizations of the matrix and testing the null hypothesis that host and virus species are randomly associated. We generated residuals for each host–virus link (smaller residuals indicate more support for coevolution, and higher residuals indicate more support for host-switching). We then visualized the host–virus links with a tanglegram of all known links in our dataset using the package 'phytools' [42].

## Results

### Overview of dataset

Our final database consists of 1,476 individual records from 118 studies of paramyxoviruses in pteropodids published between 1971 and 2023 (S1 Data; S1 Text). Each record constitutes a detection attempt associated with a unique combination of the following columns: host identification (at the finest taxonomic resolution available, typically species), virus identification (at the finest taxonomic resolution available, typically genus or species), detection type (single or pooled), study type (cross-sectional, longitudinal, or both), general detection method (PCR, serology, or viral isolation), specific detection method (i.e., type of PCR or serological assay), gene or antigen target, sample type (e.g., tissue, swab, blood, urine), type of tissue tested, country, state/region, site (with coordinates if reported), sampling year, and sampling month/season. Each record also includes the number of samples tested, the number of positive samples, and the reported prevalence.

Only three of the 118 studies were published before the year 2000, reflecting the rapid increase in interest in pteropodid–henipavirus relationships after the discovery of Hendra and Nipah virus in the 1990s (Fig 1). Of the total records, 66% (n = 972; 3.0% mean positivity) involve PCR detection attempts, 27% (n = 393; 25.8% mean positivity) involve serological detection attempts, and 7% (n = 111; 0.7% mean positivity) involve viral isolation attempts. Over 67% of records focus on the viral subfamily *Orthoparamyxovirinae*, and 66% of records involve henipaviruses specifically (the remaining 1% reflected "henipa-like" orthoparamyxoviruses). Another 11% involve the viral subfamily *Rubulavirinae*, with the remaining records attempting to detect any paramyxovirus (typically with PCR primers targeting the L gene). Detections of unclassified sequences comprise under 5% of all records. Fine-grained spatial studies of paramyxovirus prevalence are limited by available spatial metadata; only 21% of records have associated latitude and longitude listed in the original studies. However, 68% include a specific site (or multiple sites), 89% include an administrative area within a country (e.g., state, province, or district), and 99% name a specific country, allowing for broader studies of geographic biases in sampling effort (and potential georeferencing for future studies). The most common sample types in our dataset are blood (29%; n = 425), urine (27%; n = 405), swabs (22%; n = 323), and tissue (16%; n = 232). Of the blood samples, 90.8% were used for serological testing, 8.7% for PCR, and 0.5% for viral isolation. Detailed breakdowns of sampling effort and viral positivity by tissue type, detection method, and viral subfamily are included in S1 and S2 Tables.

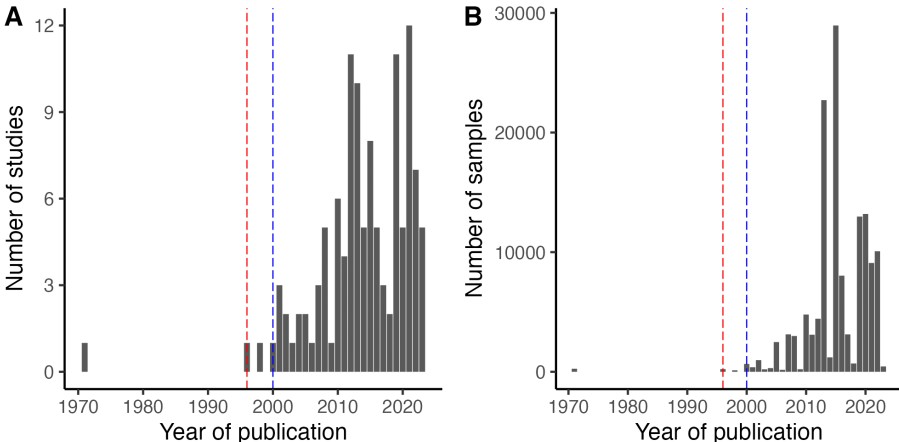

**Fig 1. Sampling effort of paramyxoviruses in Pteropodidae across time. (A)** Studies by year of publication. **(B)** Samples by year of publication (as collection year is often reported as a range of time). The vertical dashed red and blue lines indicate the first isolations of Hendra and Nipah virus, respectively, from *Pteropus* bats [8,9].

## Spatial sampling biases

Our dataset includes studies in all three regions within the Pteropodidae range (Africa, Asia, and Oceania) across 33 countries/territories (Fig 2 and S3 Table). However, sampling effort has been highly heterogeneous across countries, with the number of studies per country ranging from one to 29 (Fig 2a) and the number of samples per country ranging from 21 (Gabon) to 58,590 (Australia) (Fig 2b). In particular, large geographic sampling gaps are found in central and southern Africa (Fig 2). A generalized linear model (GLM) of binary sampling effort and subsequent post-hoc comparisons did not

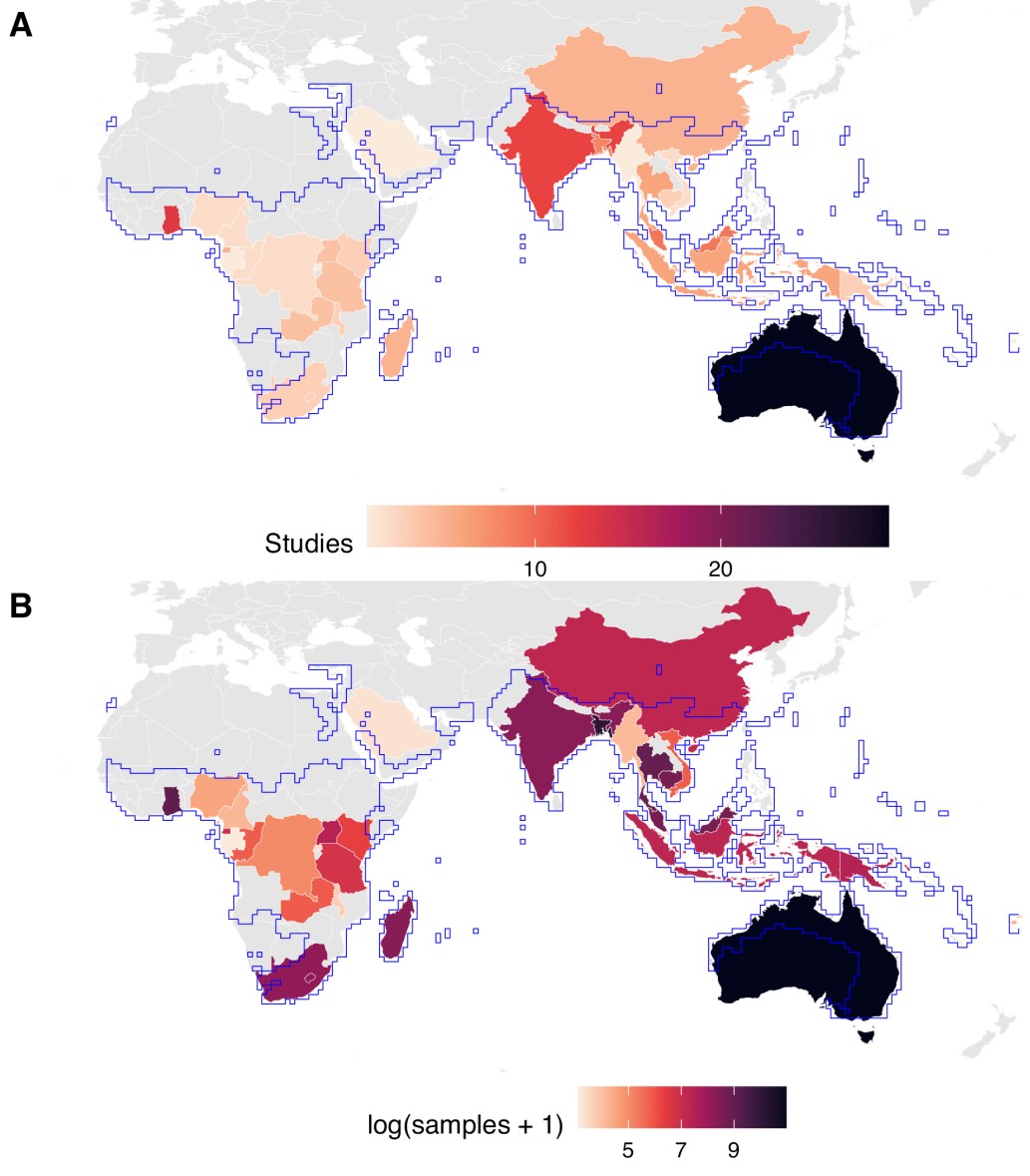

**Fig 2. Geographic distribution of paramyxovirus sampling effort in pteropodids.** The aggregated Pteropodidae IUCN distribution is outlined in blue. Note that in the UN Geoscheme, Saudi Arabia and Indonesia are classified as Asia. The base map was obtained from Natural Earth in the public domain (https://www.naturalearthdata.com/). **(A)** Number of studies per country (ranging from one to 29). **(B)** Number of samples tested per country (ranging from 21 to 58,590, presented on a log scale).

reveal any statistically significant variation across the three regions ($\chi^2 = 4.66$, $p = 0.10$, $R^2 = 0.03$) (Table 1a). However, the intensity of sampling did vary across regions, in terms of both the number of studies per country ($\chi^2 = 26.13$, $p < 0.0001$, $R^2 = 0.11$) and the number of samples per country ($\chi^2 = 147,804$, $p < 0.0001$, $R^2 = 0.38$). Specifically, there have been more pteropodid–paramyxovirus studies and samples from Oceania than from Asia or Africa (Table 1b and Table 1c), a difference likely driven by an ongoing research focus on the *Pteropus*–Hendra virus system in Australia. There have also been more samples collected from Asia than from Africa (Table 1c).

## Taxonomic sampling biases

Pteropodids sampled for paramyxoviruses include 58 of the 194 species and 23 of the 43 genera in the family. Using a phylogeny of all the Pteropodidae [36], we found intermediate phylogenetic signal in binary sampling effort ($D = 0.85$), departing from both phylogenetic randomness ($p = 0.03$) and Brownian motion models of evolution ($p = 0$). We then applied phylogenetic factorization, a graph-partitioning algorithm [38], which identified one clade with a disproportionately high fraction of sampled species, including *Rousettus amplexicaudatus*, *R. spinalatus*, *R. leschenaultii*, *R. obliviosus*, *R. madagascariensis*, *R. aegyptiacus*, *R. lanosus*, and *R. celebensis*, as well as the genera *Epomops*, *Epomophorus*, *Micropteropus*, *Nanonycteris*, *Hypsignathus*, *Megaloglossus*, and *Myonycteris* (Fig 3a and Table 2a). Among sampled species, the number of studies per species range from one to 24 and was randomly distributed across the host phylogeny (Pagel's λ = 0.00), departing from Brownian motion clustering ($p < 0.0001$) but not from phylogenetic randomness ($p = 1$). *Rousettus leschenaultii*, *R. obliviosus*, *R. madagascariensis*, and *R. aegyptiacus* are overrepresented relative to other taxa, as have been many species of *Pteropus*; however, *P. chrysoproctus*, *P. tonganus*, *P. neohibernicus*, and *P. temminckii* have been significantly understudied (Fig 3b and Table 2b). The number of samples per species range from one to 24,954 and showed moderate phylogenetic signal (λ = 0.59), departing from both Brownian motion clustering ($p < 0.0001$) and phylogenetic randomness ($p = 0.004$). The single identified overrepresented clade constitutes 16 *Pteropus* species, including those heavily sampled for Hendra and Nipah viruses (e.g., *P. giganteus*, *P. lylei*, *P. poliocephalus*, *P. alecto*) (Fig 3c and Table 2c).

On the viral side, 16 of the 78 recognized species in the family *Paramyxoviridae* [3] have been detected (by PCR, serology, or viral isolation) in the Pteropodidae, comprising three genera: *Henipavirus*, *Orthorubulavirus*, and *Pararubulavirus*.

**Table 1. Post-hoc pairwise comparisons from GLMs of sampling effort by region.**

| Comparison | Odds ratio (95% confidence interval) | z ratio | p value |
|---|---|---|---|
| 1. Binary sampling effort (binomial response) | | | |
| Africa / Asia | 1.39 (0.49-3.94) | 0.76 | 1.00 |
| Africa / Oceania | 3.71 (0.74-18.54) | 1.95 | 0.15 |
| Asia / Oceania | 2.67 (0.51-14.02) | 1.42 | 0.47 |
| B. Number of studies (Poisson response) | | | |
| Africa / Asia | 0.70 (0.45-1.09) | -1.92 | 0.17 |
| Africa / Oceania | 0.30 (0.18-0.51) | -5.51 | **<0.0001** |
| Asia / Oceania | 0.43 (0.26-0.73) | -3.84 | **0.0004** |
| C. Number of samples (Poisson response) | | | |
| Africa / Asia | 0.29 (0.28-0.29) | -151.06 | **<0.0001** |
| Africa / Oceania | 0.06 (0.06-0.06) | -355.39 | **<0.0001** |
| Asia / Oceania | 0.20 (0.20-0.21) | -262.13 | **<0.0001** |

(A) binary sampling; (B) number of studies; and (C) number of samples across countries as a function of general geographic region. The *p*-values are adjusted for multiple comparisons using a Bonferroni correction. Confidence intervals are back-transformed from the log scale.

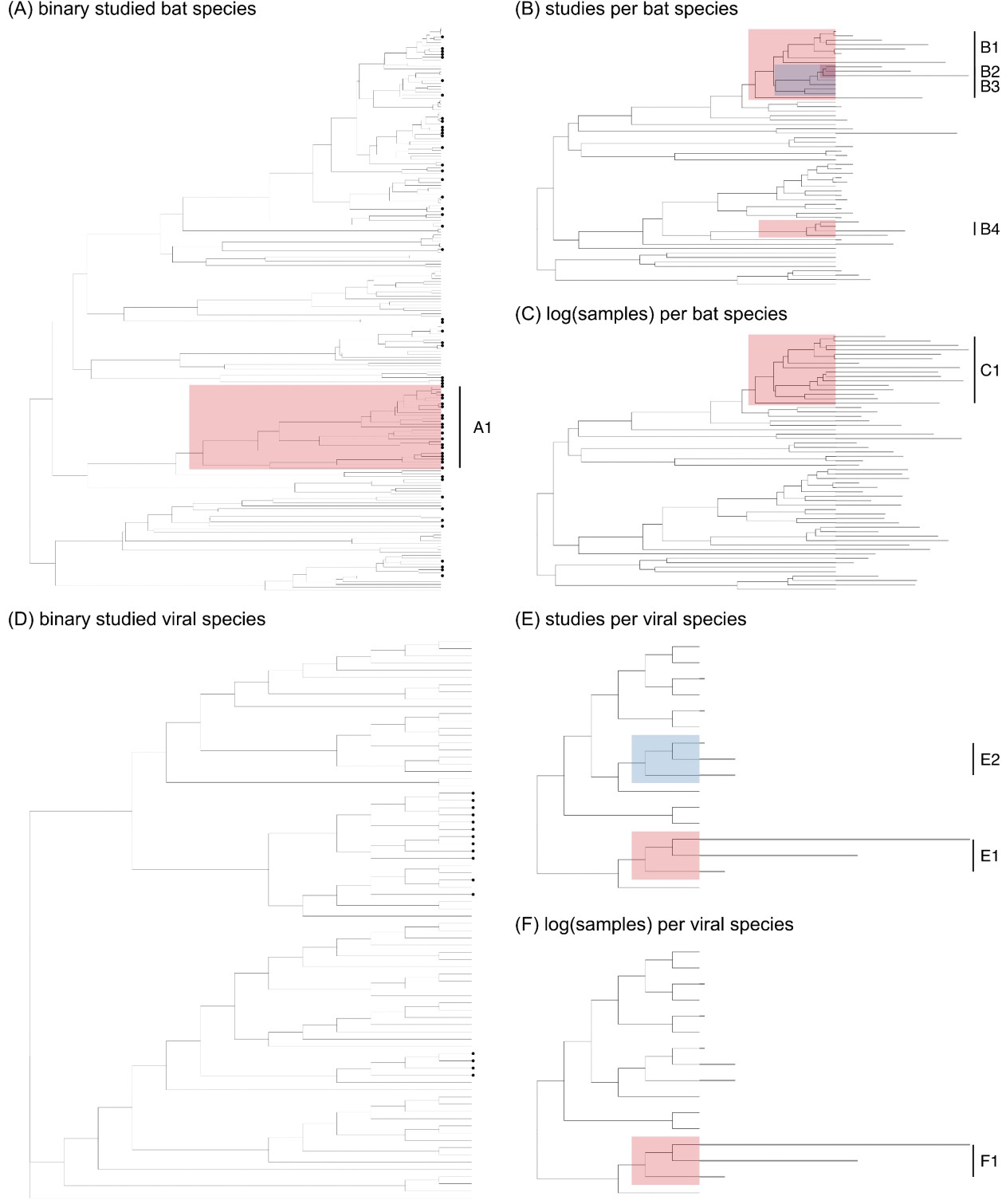

**Fig 3. Host and viral taxonomic biases in pteropodid–paramyxovirus sampling effort. (A)** Whether or not a bat species has been sampled for paramyxoviruses, across all Pteropodidae (black dots indicate species that have been sampled); **(B)** number of studies per sampled bat species; **(C)** logged number of samples per sampled bat species; **(D)** whether or not a viral species has been detected in Pteropodidae (black dots indicate viruses that have been detected); **(E)** number of studies screening for each detected viral species; **(F)** logged number of samples screened for each detected

viral species. Red and blue shading indicates clades with greater or lesser sampling effort, respectively, in comparison to all other taxa (as identified by phylogenetic factorization), with clade numbers corresponding to Table 2. In plots **(B)**, **(C)**, **(E)**, and **(F)**, segment length represents relative sampling effort.

**Table 2. Bat and virus clades differing significantly from the rest of their respective trees in terms of different measures of sampling effort, as identified by phylogenetic factorization.**

| Clade | Taxa | Number of species | Clade mean | Other mean |
|---|---|---|---|---|
| A) Binary sampling of Pteropodidae | | | | |
| A1 | *Rousettus amplexicaudatus, Rousettus spinalatus, Rousettus leschenaultii, Rousettus obliviosus, Rousettus madagascariensis, Rousettus aegyptiacus, Epomops, Epomophorus, Micropteropus, Nanonycteris, Hypsignathus, Megaloglossus, Myonycteris, Rousettus lanosus, Rousettus celebensis* | 29 | 0.62 | 0.24 |
| B) Number of studies of sampled pteropodids | | | | |
| B1 | *Pteropus chrysoproctus, Pteropus scapulatus, Pteropus vampyrus, Pteropus giganteus, Pteropus lylei, Pteropus seychellensis, Pteropus rufus, Pteropus poliocephalus, Pteropus hypomelanus, Pteropus conspicillatus, Pteropus alecto, Pteropus tonganus, Pteropus neohibernicus, Pteropus melanotus, Pteropus griseus, Pteropus temminckii* | 16 | 8.38 | 3.36 |
| B2 | *Pteropus hypomelanus, Pteropus conspicillatus, Pteropus alecto* | 3 | 15.67 | 4.15 |
| B3 | *Pteropus chrysoproctus, Pteropus tonganus, Pteropus neohibernicus, Pteropus temminckii* | 4 | 1.00 | 5.02 |
| B4 | *Rousettus leschenaultii, Rousettus obliviosus, Rousettus madagascariensis, Rousettus aegyptiacus* | 4 | 7.25 | 4.56 |
| C) Logged number of samples of sampled pteropodids | | | | |
| C1 | *Pteropus chrysoproctus, Pteropus scapulatus, Pteropus vampyrus, Pteropus giganteus, Pteropus lylei, Pteropus seychellensis, Pteropus rufus, Pteropus poliocephalus, Pteropus hypomelanus, Pteropus conspicillatus, Pteropus alecto, Pteropus tonganus, Pteropus neohibernicus, Pteropus melanotus, Pteropus griseus, Pteropus temminckii* | 16 | 6.59 | 3.91 |
| D) Binary detection of paramyxoviruses in Pteropodidae | | | | |
| E) Number of studies of detected paramyxoviruses | | | | |
| E1 | *Hendra henipavirus, Nipah henipavirus, Cedar henipavirus* | 3 | 30.67 | 2.31 |
| E2 | *Tioman pararubulavirus, Teviot pararubulavirus, Menangle pararubulavirus* | 3 | 6.00 | 8.00 |
| F) Logged number of samples of detected paramyxoviruses | | | | |
| F1 | *Hendra henipavirus, Nipah henipavirus, Cedar henipavirus* | 3 | 9.75 | 5.82 |

(A) Whether or not a bat species has been sampled for paramyxoviruses, across all Pteropodidae (0: not sampled; 1: sampled); (B) number of studies per sampled bat species; (C) logged number of samples per sampled bat species; (D) whether or not a viral species has been detected in Pteropodidae (no significant clades identified); (E) number of studies screening for each detected viral species; (F) logged number of samples screened for each detected viral species (Fig 3). We report all significant clades detected through phylogenetic factorization (identified with Holm's sequentially rejective test at a family-wise error rate of 5%), the taxa and number of species within those clades, and the mean sampling effort for the clade compared to the rest of the tree. Bat taxonomy follows Upham et al. 2019 and viral taxonomy follows Walker et al. 2022.

We found strong, highly conserved phylogenetic signal in binary sampling effort ($D = -0.90$), departing from phylogenetic randomness ($p = 0$) but not from Brownian motion models of evolution ($p = 0.99$). Phylogenetic factorization did not identify any viral clades differing significantly in the fraction of detected species (Fig 3d and Table 2d). Among detected viruses, the number of studies per virus range from one to 54 and displayed strong phylogenetic signal ($\lambda = 0.93$), departing from phylogenetic randomness ($p = 0.04$) but not from Brownian motion ($p = 0.61$). Hendra, Nipah, and Cedar henipaviruses are overrepresented in the literature, while Tioman, Teviot, and Menangle pararubulaviruses have been understudied (Fig 3e and Table 2e). The number of samples per virus range from nine to 52,006 samples and also show strong phylogenetic signal ($\lambda = 1.00$), departing from phylogenetic randomness ($p = 0.04$) but not from Brownian motion ($p = 1.00$). Again, Hendra (52,006 samples), Nipah (45,906 samples), and Cedar (2,139 samples) henipaviruses are overrepresented relative to all other detected paramyxoviruses (Fig 3f and Table 2f).

## Heterogeneity and predictors of paramyxovirus positivity

We fitted two initial, intercept-only phylogenetic meta-analysis models (one for PCR prevalence and one for sero-prevalence) that included random effects of within- and between-study variation, host genus, genus-level phylogeny, and country [36,40]. These models were fitted to aggregated datasets of PCR prevalence and seroprevalence which include different methodological approaches, primers and gene targets, and other sources of variation in viral positivity. Consequently, we observed significant heterogeneity among positivity estimates in both the PCR dataset ($I^2 = 90.0\%$, $Q = 3{,}791.92$, $p < 0.0001$) and the serological dataset ($I^2 = 96.5\%$, $Q = 9{,}733.17$, $p < 0.0001$). This heterogeneity was mostly explained by between-study (65.7% PCR; 40.7% serology) and within-study (14.0% PCR; 26.5% serology) effects. Bat phylogeny explained 0% of the heterogeneity in the PCR dataset but 28.1% of the heterogeneity in the serological dataset, and other genus effects accounted for 0% of the heterogeneity in both datasets. Country effects accounted for 8.2% and 1.3% of the heterogeneity in the PCR and serological datasets, respectively. None of the fixed effects included in our models were significantly associated with paramyxovirus prevalence or seroprevalence in Pteropodidae, with all confidence intervals crossing zero and no significant post-hoc pairwise comparisons; this may be due to the limited sample size of these separate datasets (particularly serology; $n = 311$). However, in the PCR model, serum, feces, tissue, and urine appeared to have slightly higher prevalence than the intercept (blood), and African and Asian samples had slightly higher prevalence than the intercept (Oceania). Cross-sectional studies and longitudinal studies appear to yield similar PCR prevalence rates. In the serology model, African and Asian samples had slightly lower prevalence than samples from Oceania. In both models, Eidolinae, Pteropodinae, and Rousettinae subfamilies had slightly higher viral detection than the intercept (Cynopterinae), as did Macroglossusinae in the PCR model (Fig 4).

Because tissue type had to be excluded from the phylogenetic meta-analysis models due to collinearity with other predictors, we separately investigated paramyxovirus tissue tropism in Pteropodidae by calculating PCR prevalence for each tissue type. We found that PCR prevalence was highest in spleen samples (6.9%), followed by pooled tissue samples from multiple organs (6.7%), pooled fecal samples from likely multiple individuals (6.5%), pooled swabs/samples (5.4%), and pooled urine samples from likely multiple individuals (4.2%) (Fig 5). All other tissue or sample types yielded around or under 2% prevalence: reproductive organs (2.1%); urine samples from single individuals (1.7%); kidney (1.0%); oral swabs (0.8%); fecal, rectal, or anal swabs/samples (0.8%); blood or serum (0.5%); nasal swabs (0.4%); fecal samples from single individuals (0.2%); and liver (0.1%). Lung, intestine, and skin swabs all had 0% PCR prevalence.

## Cophylogenetic analysis

Lastly, we tested the dependence of the viral phylogeny on the host phylogeny using PACo [41]. This matrix included 23 known PCR or viral isolation links between 12 pteropodid species and 13 ICTV-ratified paramyxovirus species (Fig 6). Only three viral species (*Hendra henipavirus*, *Nipah henipavirus*, and *Tioman pararubulavirus*) have been identified in more than one bat species, while over half of the included bat species (7/12) have links to more than one virus (Fig 6). When compared to 5,000 randomly permuted host–virus matrices, we found no statistically significant congruence between the two phylogenies in our dataset ($m^2_{XY} = 9{,}827.11$, $p = 0.69$). Failure to reject the null hypothesis indicates a high degree of host-switching by paramyxoviruses within Pteropodidae. However, the links between *Pteropus* species and henipaviruses and some pararubulaviruses (Teviot, Tioman, Menangle) had lower residuals and therefore demonstrated more support for host–virus coevolution, as did the links between *Rousettus leschenaultii* and Tuhoko pararubulaviruses (Fig 6 and S4 Table).

## Discussion

The isolation of Hendra and Nipah viruses from flying foxes in 1996 and 2000, respectively, sparked widespread research interest in the relationship between pteropodid bats and paramyxoviruses [8,9]. We sought to aggregate all published

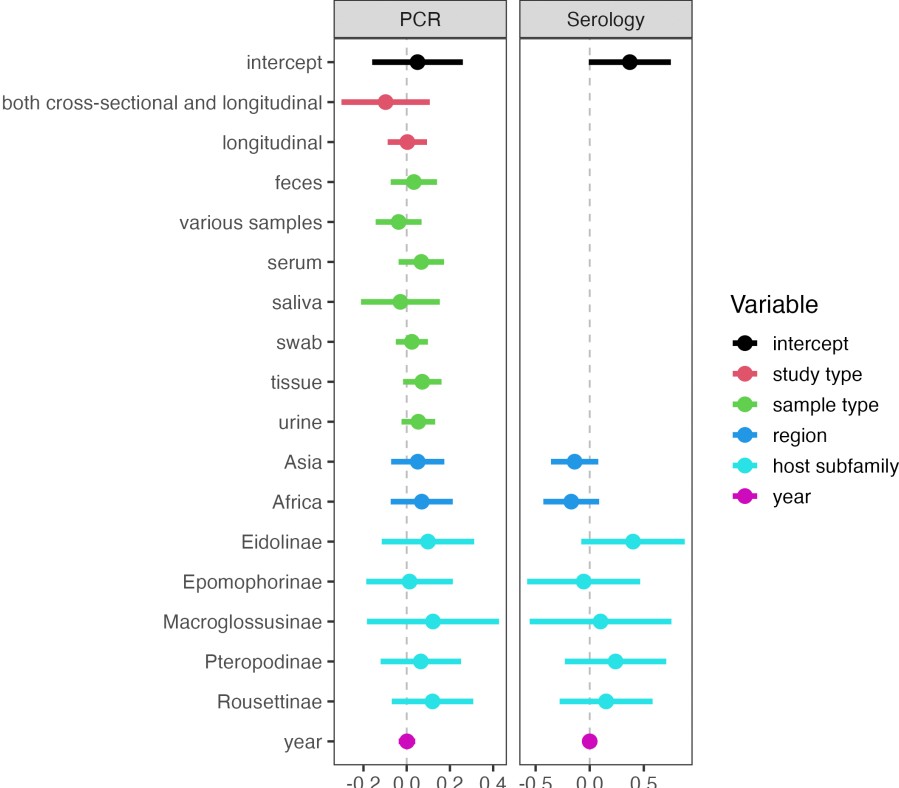

**Fig 4. Phylogenetic meta-analysis model coefficients and 95% confidence intervals, estimated with REML for each dataset.** Colours indicate the variables included in each model. Transparent confidence intervals indicate non-significant effects that cross zero (the vertical dashed line). The intercept contains the following reference levels: cross-sectional (study type); blood (sample type); region (Oceania); and Cynopterinae (host subfamily). Sample sizes are 782 PCR prevalence estimates and 311 seroprevalence estimates.

records of pteropodid–paramyxovirus associations and prevalence into a PRISMA-compliant, open, static database (S1 Fig), included here as a supplementary file and also available in a public GitHub repository (S1 Data; https://github.com/mayajuman/pteroparamyxo). Storing the dataset in this dynamic repository facilitates future updates and additional versions that include more recent studies. Our database, named pteroparamyxo, includes 1,476 records of viral detection attempts from 118 studies published between 1971 and 2023, which allows for formal synthesis of sampling and positivity across space, time, and host/viral taxonomy. This consolidation is critical for addressing questions about changes in viral dynamics [26]. Our focus on gathering detailed metadata for a viral and host family-specific database will enable robust downstream models to be trained on this taxonomic subset, an approach with higher predictive capacity than broader models [43].

Spatial and taxonomic analyses of this dataset confirm suspected trends and biases in sampling effort. Pteropodids in Australia and, to a lesser degree, South Asia (particularly India and Bangladesh) have been heavily sampled for paramyxoviruses, in terms of both number of studies and number of samples collected (Fig 2). These trends are likely driven by studies of Hendra and Nipah virus spillover events and high numbers of under-roost urine samples collected following outbreaks [44,45]. Much of Africa and Asia remains understudied for paramyxoviruses in fruit bats, despite evidence of henipavirus spillover into human populations in Cameroon, for example [46], a country which remains undersampled in our dataset (Fig 2). This sampling gap is also notable in light of the fact that African samples had slightly higher

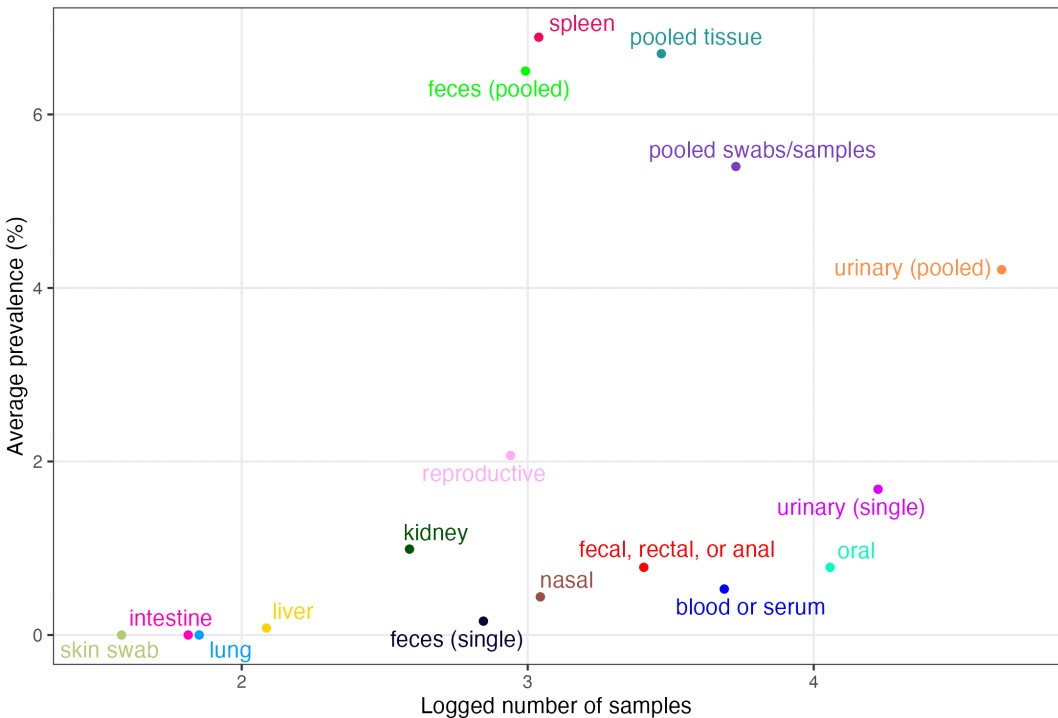

**Fig 5. PCR prevalence of paramyxoviruses in different pteropodid tissues.** Plotted by relative sampling effort (represented by the log$_{10}$-transformed number of samples screened per tissue type).

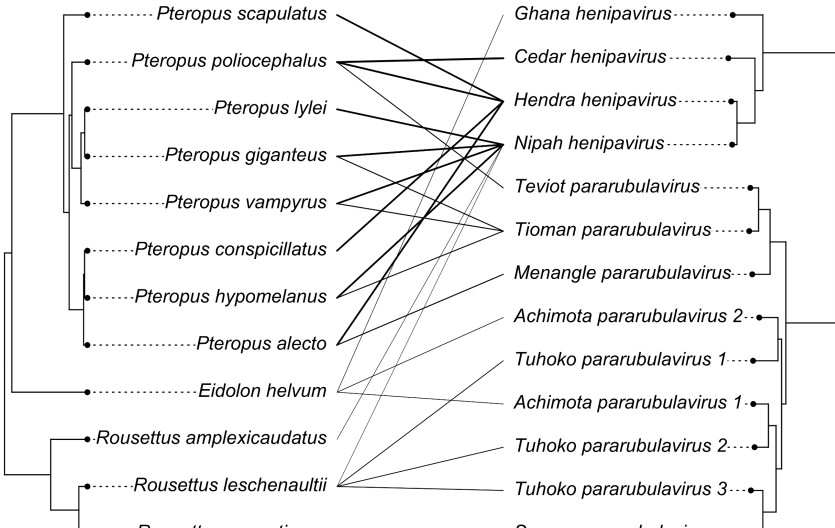

**Fig 6. Tanglegram indicating known links (PCR detections or viral isolations) between pteropodid species and ICTV-recognized paramyxovirus species.** Darker links signify smaller residuals from PACo (suggesting more support for coevolution), while lighter links signify higher residuals (more support for host-switching) (S4 Table). The phylogeny on the left indicates evolutionary relationships among pteropodid species in this dataset, with *Eidolon helvum* and the *Pteropus* genus forming a sister clade to *Rousettus* species. The phylogeny on the right indicates evolutionary relationships among viral species in this dataset, with henipaviruses and pararubulaviruses forming two distinct clades.

PCR prevalence than samples from Oceania, and some African bat subfamilies (Eidolinae and Rousettinae) had slightly higher PCR prevalence and seroprevalence than other subfamilies (Fig 4). Taken together, this suggests missed opportunities for additional detection of paramyxoviruses in African pteropodids (Fig 4).

Similarly, taxonomic sampling biases abound on both the host and viral side. In terms of binary sampling effort, a clade containing some *Rousettus* species and closely related genera (*Epomops*, *Epomophorus*, *Micropteropus*, *Nanonycteris*, *Hypsignathus*, *Megaloglossus*, *Myonycteris*) has a disproportionately high fraction of species sampled for paramyxoviruses. Among sampled taxa, some species in the highly studied genera *Pteropus*, *Rousettus*, and *Eidolon* are significantly overrepresented relative to other species; however, certain *Pteropus* species have been understudied (e.g., largely island-dwelling species like *P. chrysoproctus*, *P. tonganus*, *P. neohibernicus*, and *P. temminckii*) (Fig 3 and Table 2). The occurrence and prevalence of paramyxoviruses in these bat species are currently unknown due to very small sample sizes, and future surveillance studies could target these taxa accordingly. Those that are particularly elusive or difficult to sample in the wild may be good candidates for museum-based viral screening, an approach which allows for efficient and taxonomically broad sampling across space and time [47]. Further, while the scope of this study was limited to Pteropodidae, collecting and consolidating viral data from non-pteropodid bats could allow for broader analyses of bat–paramyxovirus coevolution. Preliminary evidence suggests that some insectivorous bat families host paramyxovirus clades distinct from those in pteropodid bats, a finding that could be more robustly assessed with additional sampling and explicit cophylogenetic tests [4,48].

On the viral side, the majority of studies have focused on henipaviruses; consequently, Hendra, Nipah, and Cedar viruses are overrepresented, while pararubulaviruses are comparatively neglected in the literature (Fig 3 and Table 2). This skew is driven by the disproportionate focus on henipavirus detection after the discovery of Hendra and Nipah viruses in flying foxes in 1996 and 2000 [8,9]. Outside of the viruses detected in pteropodids, there is a vast diversity of other mammalian and vertebrate paramyxoviruses that do not appear in our dataset (Fig 3d), including paramyxoviruses identified in insectivorous bats [49–51]. Conversely, 70 records in our database involve unclassified paramyxovirus sequences that could appear anywhere in the viral tree. Viral taxonomy can be unstable and often lags behind host taxonomy [3]; future ICTV updates may recognize additional bat-borne paramyxovirus species. Still, based on published links between 13 recognized paramyxovirus species and 12 named pteropodid species (other isolated viruses were generally attributed to *Pteropus* spp.), we performed a cophylogenetic analysis that revealed little evidence of phylogenetic congruence between hosts and viruses, contrary to tentative hypotheses of pteropodid–paramyxovirus coevolution posed in earlier surveillance studies [16]. The strongest coevolutionary signal was found in *Pteropus–Henipavirus* systems, though this result was not statistically significant and may be an artifact caused by heavy sampling of *Pteropus* species for henipaviruses. There was more support for host-switching among other links (e.g., *Rousettus/Eidolon*–pararubulaviruses). These findings raise the possibility of paramyxovirus expansion into new pteropodid hosts and geographic areas, which may have implications for spillover into human and domestic animal populations.

Our results also hint at methodologies and sample types that might yield higher paramyxovirus positivity. Over time, there has been a shift in PCR detection methods from targeted RT-PCR to metagenomic approaches; in serological studies, neutralization assays have been supplanted by multiplexed Luminex assays and ELISA (S2 Fig). In general, urine, tissue, serum, and fecal samples appeared to have higher PCR prevalence than blood, whereas saliva and swab samples had lower prevalence (Fig 4). A more in-depth look at tissue tropism suggested that spleen and pooled feces are undersampled relative to their typical prevalence, as these tissues have the highest PCR prevalence among specified sources (i.e., excluding pooled tissue) (Fig 5 and S1 Table). These sample types could be good priorities for future surveillance studies of paramyxoviruses in pteropodids and offer specific options for both invasive (e.g., spleen extraction during necropsy) and non-invasive (e.g., under-roost fecal collection [44]) sampling approaches. Pooled samples generally show higher positivity, as they contain samples from likely multiple individuals, raising the likelihood of a positive result. Both cross-sectional and longitudinal studies yielded similar PCR prevalence (Fig 4), suggesting that both are suitable

approaches for surveillance studies and that the study type can be selected based on the research question at hand. However, only 7% of records involved viral isolation attempts, the gold standard for confirming viral presence in a sample. This is due in part to the strict biosafety requirements and resources (e.g., BSL-4 facilities) involved in viral isolation for henipaviruses, which inhibit detection attempts in countries with limited resources where these viruses are known to circulate.

Syntheses and meta-analyses such as this can be limited in scope and statistical robustness by data gaps as well as a lack of stratified and standardized metadata reported in existing studies. Our study is inherently restricted by between-study variation at a finer level than what is captured in our dataset and in many of the original studies (e.g., sampling protocols, PCR primers, laboratory protocols, etc). Consequently, our models treat all PCR and all serological detection attempts as comparable to one another (though weighted by sample size), when in reality methodologies can differ widely across studies. Where possible, researchers should report the specific tissue types, localities, sampling dates, taxonomic identifications, and other metadata included in pooled samples; we also suggest providing detection estimates at as fine-grained a level as possible to enable more detailed analysis of pteropodid–paramyxovirus associations. This will help inform sampling protocols for surveillance of wild bats and museum specimens. Other limitations of this study include redundancy across samples from the same individual bat (e.g., the screening of tissue samples from bats already found to be positive, which artificially inflates positivity estimates) as well as reporting gaps in the literature, as our analysis only includes published data.

Here, we have: (1) highlighted the major spatial and taxonomic sampling gaps in the pteropodid–paramyxovirus system; (2) identified sample types, regions, host subfamilies, and study types associated with higher prevalence or seroprevalence; and (3) found evidence of host-switching by paramyxoviruses within Pteropodidae. Importantly, we found no evidence that paramyxovirus prevalence or seroprevalence in pteropodids has changed in recent decades, as collection year was not a significant predictor in our models (Fig 4). Our published dataset constitutes a first step toward gathering, standardizing, and stratifying surveillance data to guide future studies of paramyxoviruses, which continue to pose threats to human health. Addressing the sampling gaps outlined here will be critical in developing a more complete picture of these bat–virus systems.

## Supporting Information

**S1 Fig. PRISMA reporting diagram for study selection and screening.**
(TIFF)

**S2 Fig. Stacked barplots showing changes in diagnostic methods for (A) PCR and (B) serological detection of paramyxoviruses in Pteropodidae by publication year.**
(TIFF)

**S3 Fig. Temporal autocorrelation functions for (A) PCR prevalence and (B) seroprevalence of paramyxoviruses in Pteropodidae.** Lags represent years between estimates.
(TIFF)

**S1 Table. Database breakdown by sample type and detection method.** Sample size (records and number of samples), percentage of records with at least one positive sample, and average prevalence by detection method, sample type, and viral subfamily.
(XLSX)

**S2 Table. Database breakdown by type of paramyxovirus.** Number of studies, host species, and samples tested for different paramyxoviruses, with positivity rates reported for serological and PCR detection attempts as well as number of successful viral isolations.
(XLSX)

**S3 Table. Database breakdown by country.** Number of studies, number of samples, and mean viral prevalence and seropositivity rates for all countries.
(XLSX)

**S4 Table. Results from PACo cophylogenetic analysis of paramyxoviruses in Pteropodidae.** Jackknifed squared residuals and upper 95% confidence intervals for each known pteropodid–paramyxovirus link from PACo.
(XLSX)

**S1 Text. Complete reference list for all 118 studies included in the dataset.**
(PDF)

**S1 Data. Database generated and used in this study ('pteroparamyxo').**
(CSV)

## Acknowledgments

We thank Kim Halpin and Mohammed Ziaur Rahman for constructive feedback on earlier versions of this manuscript.

## Author contributions

**Conceptualization:** Maya M. Juman, Olivier Restif, Daniel J. Becker.

**Data curation:** Maya M. Juman.

**Formal analysis:** Maya M. Juman.

**Methodology:** Daniel J. Becker.

**Supervision:** Daniel J. Becker.

**Visualization:** Maya M. Juman.

**Writing – original draft:** Maya M. Juman.

**Writing – review & editing:** Olivier Restif, Daniel J. Becker.

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
