## [Decision Letter · Decision Letter 0]

20 Oct 2025

PNTD-D-25-01418

Paramyxoviruses in Old World fruit bats (Pteropodidae): an open database and synthesis of sampling effort, viral positivity, and coevolution

Dear Dr. Juman,

Thank you for submitting your manuscript to PLOS Neglected Tropical Diseases. After careful consideration, we feel that it has merit but does not fully meet PLOS Neglected Tropical Diseases's publication criteria as it currently stands. Therefore, we invite you to submit a revised version of the manuscript that addresses the points raised during the review process.

Please submit your revised manuscript within 30 days Dec 19 2025 11:59PM. If you will need more time than this to complete your revisions, please reply to this message or contact the journal office at plosntds@plos.org. Please include the following items when submitting your revised manuscript:

We look forward to receiving your revised manuscript.

Kind regards,

Tony Schountz, PhD

Academic Editor

Andrea Marzi

Section Editor 

Shaden Kamhawi

co-Editor-in-Chief

Paul Brindley

co-Editor-in-Chief

**Additional Editor Comments: **

Dear Dr. Juman, the reviewers have identified several minor issues with the manuscript that should be addressed. Reviewer 2, in particular, has raised several points for clarification. Please respond to each of these points for the resubmission.

**Journal Requirements:**

1) Please upload all main figures as separate Figure files in .tif or .eps format. For more information about how to convert and format your figure files please see our guidelines: 

2) Some material included in your submission may be copyrighted. According to PLOSu2019s copyright policy, authors who use figures or other material (e.g., graphics, clipart, maps) from another author or copyright holder must demonstrate or obtain permission to publish this material under the Creative Commons Attribution 4.0 International (CC BY 4.0) License used by PLOS journals. Please closely review the details of PLOSu2019s copyright requirements here: PLOS Licenses and Copyright. If you need to request permissions from a copyright holder, you may use PLOS's Copyright Content Permission form.

Potential Copyright Issues:

i) Figure 2. Please (a) provide a direct link to the base layer of the map (i.e., the country or region border shape) and ensure this is also included in the figure legend; and (b) provide a link to the terms of use / license information for the base layer image or shapefile. We cannot publish proprietary or copyrighted maps (e.g. Google Maps, Mapquest) and the terms of use for your map base layer must be compatible with our CC BY 4.0 license.

3)  Please ensure that the funders and grant numbers match between the Financial Disclosure field and the Funding Information tab in your submission form. Note that the funders must be provided in the same order in both places as well.  

**Reviewers' comments: **

Reviewer's Responses to Questions

**Key Review Criteria Required for Acceptance?**

**Methods**

-Are the objectives of the study clearly articulated with a clear testable hypothesis stated?

-Is the study design appropriate to address the stated objectives?

-Is the population clearly described and appropriate for the hypothesis being tested?

-Is the sample size sufficient to ensure adequate power to address the hypothesis being tested?

-Were correct statistical analysis used to support conclusions?

-Are there concerns about ethical or regulatory requirements being met?

Reviewer #1: Great

Reviewer #2: Minor Revision

1. Author should provide the total number of samples included for each testing method (PCR, ELISA, and virus isolation). To improve readability, include explicit numbers and positivity rates in the main text for each method and sample type (e.g., n=XX, YY% positive). (Line 255)

2. You mention that PCR prevalence and seroprevalence were not significantly influenced by the year of collection, suggesting little change across decades. I recommend to provide a sensitivity analysis to test whether methodological differences across time might obscure temporal trends? (Line 546)

**Results**

-Does the analysis presented match the analysis plan?

-Are the results clearly and completely presented?

-Are the figures (Tables, Images) of sufficient quality for clarity?

Reviewer #1: Yes - all good

Reviewer #2: Minor Revision

1. While pooled swab positivity is mentioned (5.4%) and detailed in supplementary data, the main text does not specify overall positivity or breakdown of sample types with positivity (e.g., oral or rectal swab). Add this information to the results for completeness, as it could inform non-invasive sampling strategies. (Lines 430–431)

2. The meta-analysis synthesizes data from studies spanning 1971–2023, but study periods are not explicitly included as a factor. Please add this to the methods and results sections, as temporal clustering of studies could influence overall prevalence estimates due to varying environmental or methodological factors.

**Conclusions**

-Are the conclusions supported by the data presented?

-Are the limitations of analysis clearly described?

-Do the authors discuss how these data can be helpful to advance our understanding of the topic under study?

-Is public health relevance addressed?

Reviewer #1: Yes

Reviewer #2: Revision required as per recommendations

1. Your findings highlight that pararubulaviruses are understudied compared to henipaviruses. Could you elaborate on why this bias exists in details? (Line 511)

2. Since PCR prevalence and seroprevalence represent different methodologies, so require more explanation to avoid potential confounding and better highlight method-specific patterns. Explain it in the results and discussion section.

3. The finding of higher prevalence in spleen samples (6.89%) is highlighted (Figure 5) may be affected by redundancy, since individuals positive in other tissues (e.g., swabs) could bias detection rates, especially given the small sample size. Could you clarify this point and discuss the limitations of sample size in tissue-specific conclusions?

4. I recommend adding a separate paragraph explicitly outlining the study’s limitations (e.g., heterogeneous methods across studies, reporting gaps, sample redundancy, uneven sample sizes, geographic under-sampling in Africa, reliance on published data only).

5. Virus isolation is notably low compare to PCR, likely due to requirements for BSL-4 facilities that is very limited worldwide, especially for henipaviruses. Expand on this in the discussion section to explain why PCR dominates and implications for confirming viable virus.

6. Pooled urine and fecal samples from roosts show higher positivity, possibly because they mix contributions from multiple bats. Discuss this in the results and discussion section to guide interpretation for colony-based surveillance.

**Editorial and Data Presentation Modifications?**

Reviewer #1: Reference 3 does need to be changed to Walker et al (as is indicated in the author's response).

Reviewer #2: Accept

**Summary and General Comments**

Reviewer #1: The authors have clearly addressed all 3 reviewer's comments and suggestions.

There was only one minor change that was not done - and that was updating reference 3 to Walker et al.

I also want to commend the authors for their considered responses to Reviewer 1 who was unprofessional and dismissive in their approach to providing constructive feedback.

Reviewer #2: The paper is clearly written, conceptually sound, and well-organized. The results are important for understanding the ecology of paramyxoviruses in pteropodid bats and for directing future monitoring activities. The public repository (GitHub) is an important resource for the scientific community. However, the manuscript has a few points for improvement to enhance clarity.

PLOS authors have the option to publish the peer review history of their article (what does this mean?). If published, this will include your full peer review and any attached files.

Reviewer #1: **Yes:** Kim Halpin

Reviewer #2: **Yes:** Mohammed Ziaur Rahman

**Figure resubmission:**

After uploading your figures to PLOS’s NAAS tool - https://ngplosjournals.pagemajik.ai/artanalysis, NAAS will process the files provided and display the results in the 'Uploaded Files' section of the page as the processing is complete. If the uploaded figures meet our requirements (or NAAS is able to fix the files to meet our requirements), the figure will be marked as 'fixed' above. If NAAS is unable to fix the files, a red 'failed' label will appear above. When NAAS has confirmed that the figure files meet our requirements, please download the file via the download option, and include these NAAS processed figure files when submitting your revised manuscript.
---

## [Editor Report · Decision Letter 1]

29 Oct 2025

Dear Ms. Juman,

We are pleased to inform you that your manuscript 'Paramyxoviruses in Old World fruit bats (Pteropodidae): an open database and synthesis of sampling effort, viral positivity, and coevolution' has been provisionally accepted for publication in PLOS Neglected Tropical Diseases.

Best regards,

Tony Schountz, PhD

Academic Editor

Andrea Marzi

Section Editor

Shaden Kamhawi

co-Editor-in-Chief

Paul Brindley

co-Editor-in-Chief

Dear Dr. Juman,

Thank you for addressing the reviewer comments. I have recommended acceptance of your manuscript.

---

## [Editor Report · Acceptance letter]

Dear Ms. Juman,

We are delighted to inform you that your manuscript, "Paramyxoviruses in Old World fruit bats (Pteropodidae): an open database and synthesis of sampling effort, viral positivity, and coevolution," has been formally accepted for publication in PLOS Neglected Tropical Diseases.

Best regards,

Shaden Kamhawi

co-Editor-in-Chief

Paul Brindley

co-Editor-in-Chief
